# Prevalence and Demographic Correlates of Substance Use among Adults with Mental Illness in Eastern Cape, South Africa: A Cross-Sectional Study

**DOI:** 10.3390/ijerph18105428

**Published:** 2021-05-19

**Authors:** Linda Tindimwebwa, Anthony Idowu Ajayi, Oladele Vincent Adeniyi

**Affiliations:** 1Department of Psychiatry, Faculty of Health Sciences, Walter Sisulu University/Cecilia Makiwane Hospital, East London 5200, South Africa; 2Population Dynamics and Sexual and Reproductive Health, African Population and Health Research Centre, APHRC Campus, Manga Close, Nairobi 00100, Kenya; aajayi@aphrc.org; 3Department of Family Medicine, Faculty of Health Sciences, Walter Sisulu University/Cecilia Makiwane Hospital, East London 5200, South Africa; oadeniyi@wsu.ac.za

**Keywords:** alcohol use, Eastern Cape, psychoactive drugs, South Africa, substance use

## Abstract

This study reports on the prevalence and demographic correlates of substance use among individuals with mental illness in the Eastern Cape, South Africa. This cross-sectional study was conducted in the Outpatient Clinic of a large hospital in the Eastern Cape, South Africa. A pre-validated tool on alcohol and psychoactive drug use was administered to 390 individuals with mental illness. Multivariable logistic regression models were fitted to explore the demographic correlates of alcohol and psychoactive drug use. Of the total participants (N = 390), 64.4% and 33.3% reported lifetime (ever used) and past-year use of alcohol, respectively, but the prevalence of risky alcohol use was 18.5%. After adjusting for relevant covariates, only male sex, younger age, and rural residence remained significantly associated with risky alcohol use. The prevalence of ever-use and past-year use of psychoactive substances was 39.7% and 17.4%, respectively. The most common substance ever used was cannabis (37.4%). Male sex, younger age, owning a business, and being unemployed were significantly associated with higher odds of lifetime and past-year use of psychoactive substances. Findings highlight the need for dedicated infrastructure and staff training in the management of these dual diagnoses in the region.

## 1. Introduction

Substance use is an important risk factor for morbidity, disability, and premature death globally [1]. Besides the adverse consequences on health, alcohol and psychoactive drugs also create a significant burden on productivity and economic and social aspects of individuals, as well as their families and communities [2]. It was estimated that in 2017, 271 million people (5%) of the global population between 15 and 64 years of age had used psychoactive drugs in the previous twelve months [3]. A national survey of South African adults showed a prevalence of 33.1% for past-year alcohol use, which was higher than the global average of 32.5% [4,5]. In addition, a total of 62,300 alcohol-related deaths were reported in 2015 [6].

Several studies have reported a higher prevalence (range of 21.3–55.6%) of substance use disorders in individuals with mental illness compared to three to four percent reported in the general population [7,8]. In Nigeria, two studies on the use of psychoactive substances in mentally ill individuals showed prevalences between 21.3% and 29.3% [9,10]. In Finland, Karpov et al. (2017) [11] reported a prevalence of 27.5%. The Epidemiologic Catchment Area Study, a large study carried out in the United States, found a prevalence rate of co-morbidity of 29% [12]. A study done in the Western Cape of South Africa showed a prevalence rate of 55.6% [13]. All of the studies had a confidence interval of 95%.

Substance use and mental illness are strongly inter-related [14,15]. Several theories have been advanced to explain the bi-directional and complex relationship between these dual diagnoses, which often co-exist in the same individual. The self-medication hypothesis postulates that mental illness might contribute to substance use and addiction due to individuals using specific substances to ameliorate specific symptoms [16,17,18,19]. Another theory suggests that individuals with harmful substance use have a higher risk of developing a psychiatric disorder [16,18,20]. For psychiatric disorders to evolve in individuals with a history of substance use, the authors of [21] argued that genetic predisposition or environmental factors have crucial roles to play. It has also been hypothesized that substance abuse and psychiatric disorders originate from a common, overlapping genetic vulnerability in the affected individuals [16,18]. Bi-directional models suggest that either disorder increases the vulnerability to the other in the form of a feedback loop [18,20].

The complex relationship between substance use and mental illness negatively impacts the natural course, clinical outcomes, and prognosis of either condition [22,23]. Substance use in individuals with mental illness leads to poor adherence to treatment, deterioration of condition, recurrent relapses, and hospitalizations [22,23]. Individuals with co-morbidity of substance and psychiatric disorders are more likely to be unemployed, homeless, and violent and have family and interpersonal relationship instability [22,23]. They tend to experience victimization, social exclusion, legal problems, and a greater prevalence of acquiring communicable diseases like HIV or Hepatitis, due to their risk-taking behaviors [24,25,26].

Cannabis is the most commonly used psychoactive drug globally. In 2017, an estimated 188 million (3.8%) of the global population between 15 and 64 years of age had used cannabis in the previous year [3]. The term opioids include both opiates and non-medical use of prescription opioids. Examples of these are heroin, morphine, codeine, tramadol, and fentanyl. Past year opioid use in the same age group globally in 2017 was estimated at 53.4 million (1.1%), an increase of 56% from 2016. Amphetamines, methamphetamine, in particular, had an estimated 28.9 million (0.6%) past year users globally [3]. A national survey in South Africa looking at the past 3-month use of psychoactive drugs found that cannabis (dagga, weed, zol) was the most used drug with a prevalence of 4.0%, which is higher than the global average. Following cannabis were sedative-hypnotics, examples of which are diazepam (valium), methaqualone (mandrax) and Rohypnol (0.4%). Amphetamines (tik, speed), cocaine (coke, crack, rock), and opiates each had a prevalence of use of 0.3% [8]. Authors [27] Thungana et al. (2019) reported a prevalence of 81.9% among individuals admitted with first-episode psychosis (FEP) in the Nelson Mandela Bay of Eastern Cape Province. Findings of Thungana et al. (2019) are limited to people with FEP and do not reflect the broader categories of mental illnesses managed at the province’s health facilities.

Previous studies on alcohol use and harmful drinking in the Buffalo City Municipality and Amathole district in South Africa have focused on the general population [28,29], neglecting individuals with mental illness. This study reports on the prevalence and correlates of substance use (risky use of alcohol and psychoactive drugs) among individuals with mental illness attending the Outpatient Clinic of the Cecilia Makiwane Hospital Mental Health Unit (CMHMHU) in East London, Eastern Cape, South Africa. The study’s findings could guide the crafting of context-specific interventions and policies on the prevention and continued support for individuals with dual diagnoses of mental illness and substance use.

## 2. Materials and Methods

### 2.1. Study Design and Setting

This cross-sectional study was conducted in the Outpatient Clinic of the Cecilia Makiwane Hospital Mental Health Unit (CMHMHU) situated in Mdantsane, East London, between March and June 2020. This department provides both inpatient and outpatient mental health care to the 1,674,637 people living in Buffalo City Metropolitan Municipality (BCMM) and Amathole district in the Eastern Cape Province of South Africa [30]. The department is an academic unit affiliated with Walter Sisulu University, South Africa, for the training of undergraduate and postgraduate students in psychiatry. The inpatient facility consists of 50 beds, 25 male and 25 female. The outpatient unit sees between 13,000 and 15,000 patients a year.

### 2.2. Participants and Sample Size

The sample size for this study was estimated as 390, using Cochran’s formula for cross-sectional study [31]:N = Z1 − ∝/22p (1 − p)/d^2^(1)
where z1 − ∝/2 is the standard normal variate at 1.96.

*p* = expected proportion of patients with co-morbidity of substance use and mental illness, based on a previous study [32], and observation of patients admitted in the wards at CMHMHU, at 50%, which is equal to 0.5. D is the absolute precision and is taken as 0.05.

All individuals attending CMHMHU outpatients’ clinic were considered eligible for the study. Participants were included if they were 18 years and older, had been on treatment for mental illness for at least a year, and had no violent or aggressive behavior. However, some participants were excluded if they displayed behavior that posed a danger to themselves, others, and property during the study. Such individuals were immediately offered emergency care per departmental protocol.

### 2.3. Study Instrument and Pilot Study

A questionnaire was designed using components of the validated National Insitute on Drug Abuse (NIDA), Rockville, MD, United States Quick Screen [33] and WHO STEPwise questionnaires [34]. The study instrument contained three sections: demographic data, use of alcohol and psychoactive drugs, and the current diagnosis of mental illness. The tool was piloted with ten patients at the study site in order to ascertain its acceptability and feasibility. Using feedback from the participants and the attending clinicians, final adjustments were made to the questionnaire. Participants of the pilot were not included in the main study.

### 2.4. Study Procedure

Information about the study, its purpose, and the questionnaire were provided to participants verbally and through an information sheet. Doctors were given information sheets regarding the eligibility criteria and trained on how to administer the questionnaire, which was completed after consultation with the participants. To supplement the information obtained from self-report by the participants, medical records were reviewed in order to validate the responses given. Where discrepancies were found, the participants’ clinical notes were taken as the true reflection of the use of the substances under inquiry. Additionally, collateral information was used to validate the self-report of the main outcome measures. Participants were recruited consecutively by the attending doctors. Data collection was done over a period of four months, from March to June 2020. Given that this study was conducted during the COVID-19 pandemic, we strictly followed the necessary precautions based on local protocols. 

### 2.5. Measures

#### 2.5.1. Outcome Variables

Alcohol use was measured in terms of a “standard drink” in order to classify use as risky or non-risky. There is no global consensus for what constitutes a “standard drink”, with a study finding significant variations between countries [35]. The World Health Organisation (WHO), Geneva, Switzerland, adopted ten grams of pure alcohol per drink as the definition of a standard drink. In South Africa, one unit of alcohol is twelve grams of pure alcohol per drink. Low-risk drinking is classified as twenty-four grams (2 units) daily for both men and women [35].

Using the definition by the National Institute on Alcohol Abuse and Alcoholism, 2020, risky alcohol use in this study was defined as consumption of four or more standard drinks by men and three or more standard drinks by women daily, or, on a weekly basis, 21 drinks for men and 14 drinks for women. Alcohol use was self-reported and categorized as current use if participants had consumed alcohol in the past twelve months (or past year) [36] and ever-use (lifetime).

Psychoactive drugs included in this study were those that could result in dependence and whose use was deemed risky. The selection of substances was informed by the commonly used psychoactive drugs in South Africa reported by [37]. These include dried herbal cannabis (dagga), methaqualone (mandrax), crystal methamphetamine (tik), opioids, and cocaine [37]. In order to accommodate outliers, the category of “others” was included. Use of psychoactive substances was elicited by self-report and categorized as past year use and ever-use (lifetime).

#### 2.5.2. Covariates

Pertinent demographic characteristics considered in the study, age, sex, highest level of education, race, marital status, employment, monthly household income, and area of residence (rural vs. urban) were obtained by self-reporting. The participants’ ages (in years) were grouped as 18–25, 26–35, 36–45, 46–55, 56–65, and 66 years or older. Their highest level of education was grouped according to grades, with any post-high school education categorized as tertiary. Household monthly income in rands was categorized as none, R150–2000, R2001–5000, and R5001 or more. Employment status was categorized as none, having a job or business, and social assistance (child support grant, older person’s grant, disability grant, grant-in-aid, care dependency grant, war veteran’s grant, foster child grant and social distress relief grant). Participants were considered unemployed if they had no work in both the formal or informal sectors.

### 2.6. Ethical Considerations

Ethical approval for the study was received from the Walter Sisulu University Research Ethics Committee (reference number 093/2019). Also, the Eastern Cape Department of Health and the clinical governance of Cecilia Makiwane Hospital, Mdatsane, South Africa, granted permission for the researcher to conduct the study. Participants and/or their legal guardians, in the event that the patient was unable to give consent due to diminished capacity, received an information sheet in their preferred language (English or Xhosa) detailing the study’s purpose and process. Participants (or guardians) who were willing to participate in the study gave informed consent in writing. We respected participants’ rights to privacy and confidentiality of medical information during and after the study. All soft data related to the survey were password-protected, while hard copies of materials were kept under lock and key. The study followed the Helsinki Declaration on human and animal research guidelines.

### 2.7. Statistical Analysis

Data were entered into an Excel spreadsheet, and analysis was conducted using the IBM Statistical Package for Social Sciences Version 24.0 for Windows (IBM Corp., Armonk, NY, USA). Percentages (%) and frequencies (n) were used to report categorical variables, with continuous variables being reported as means. We used simple descriptive statistics to summarize the participants’ socio-demographic characteristics and bivariate analysis to examine the associations between background characteristics and alcohol and psychoactive drug use. We used multivariate logistic regression models to explore the demographic correlates of alcohol and psychoactive drug use. A *p*-value of <0.05 was considered to be statistically significant. We tested for collinearity using variance inflation factors (VIF), and none of the variables had a VIF of more than 2, indicating that there was no multicollinearity.

## 3. Results

### 3.1. Demographic Characteristics of the Participants

Most of the participants were male (59.5%), single (74.9%), Black (87.2%), aged 35 years and above (68.2%), and had attained an education level of Grade 8 to 12 (57.2%). The majority of the participants were unemployed (81%). However, 67.4% (n = 263) of the total participants received social assistance from the government in the form of disability, pensions, and child-care grants. Approximately 47% of the participants (n = 183) reported a combined household income of R2001–R5000 per month. Most of the participants lived in urban areas (83.6%) with their families (84.9%) (Table 1). Schizophrenia was the predominant mental illness diagnosed among the participants (40.8%), followed by cannabis-related disorders (19.2%) and major depressive illness (18.5%) (Appendix A).

### 3.2. Patterns of Alcohol Use and Their Predictors

Of the total participants (N = 390), 64.4% and 33.3% reported ever-use and past-year use of alcohol, respectively (Table 2). The prevalence of ever-use and past-year use of alcohol was higher among males (77.2% and 37.9%, respectively) compared to females (45.6% and 26.6%, respectively). The prevalence of past-year and ever use of alcohol reduced with an increase in age. In the unadjusted logistic regression model, male sex, younger age (<56 years), source of income, and having a lower level of education (<grade 8) were significantly associated with ever-used alcohol. However, after adjusting for relevant covariates, only male sex and younger age remained significantly associated with higher odds of having ever used alcohol, while having a lower level of education (<grade 8) was associated with lower odds of having ever used alcohol.

Regarding current use from the unadjusted analysis, male sex, younger age, being unemployed, owning a business, lower level of education (<grade 8), and urban residence was significantly associated with past-year alcohol use. The direction and magnitude of the associations remained in the adjusted model, except for the source of income. Males were twice more likely to have used alcohol in the past year in comparison to females. Individuals aged 18–35 years were over three times more likely to have past-year use of alcohol when compared to those aged 56 years and above. Individuals with a lower level of education (<grade 8) and those living in rural areas had lower odds of past-year alcohol use (Table 2). No significant relationship existed between the type of underlying mental illness and risky alcohol use; however, bipolar and related disorders were significantly associated with past-year psychoactive drug use (Appendix A).

The prevalence of risky alcohol use was 18.5%. The prevalence of risky alcohol use was higher among males (24.6%), those aged 18–35 (25.8%) and 46–55 (19.8%), and those living in urban areas (18.8%) compared with females (9.5%), aged 56 and above (7.1%), and those living in rural areas (6.7%). In the unadjusted regression model, male sex, younger age (18–35 years), and unemployment were significantly associated with higher odds of risky alcohol use. Rural residence was associated with lower odds of risky alcohol use. After adjusting for relevant covariates, only male sex, younger age, and rural residence remained significant, with the direction and magnitude remaining unchanged (Table 3).

### 3.3. Psychoactive Substance Use

The most common substance ever used was cannabis (37.4%), followed by methaqualone (11%), methamphetamines (10.3%), inappropriate OTC/prescription drugs (3.1%), cocaine (3.1%), and opioids (3.1%). (Table 4).

### 3.4. Predictors of Psychoactive Substance Use

The prevalence of ever-use and past-year use of psychoactive substances was 39.7% and 17.4%, respectively. In the unadjusted regression model, male sex, younger age (18–35 years), owning a business, and being unemployed were significantly associated with higher odds of having ever used psychoactive substances. After controlling for relevant covariates, the magnitude and direction of effects remained. Males were approximately eight times more likely to have ever used psychoactive substances than females. Those aged 18–35 years were six times more likely to have ever used psychoactive drugs than individuals aged 56 years and above. Relative to those receiving social assistance, individuals who were employed or owned a business were five times more likely to report having ever used psychoactive drugs. Marital status became significant after adjustment, with married or cohabiting participants having a higher likelihood of lifetime substance use than their single counterparts.

Similar findings emerged regarding psychoactive substance use in the past year. Males were approximately eight times more likely to have used psychoactive substances in the past year compared to females. Individuals aged 18–35 years had a higher likelihood of past-year psychoactive substance use than those 56 years and above after adjusting for other covariates. Individuals living in rural areas were 73% less likely to have used any psychoactive drugs in the past year compared to those living in urban areas. Those who were employed were more likely to have used drugs in the past year in comparison to those who were receiving social assistance (grants) from the government (Table 5).

## 4. Discussion

Several studies have examined alcohol and psychoactive drug use patterns in different population groups in South Africa [38,39,40,41], but individuals with mental illness rarely receive attention. As such, relevant data in this vulnerable population are not available to guide policy development and craft appropriate interventions. Therefore, this study addresses this gap by examining the prevalence and demographic correlates of alcohol and psychoactive drug use among individuals with mental illness in the central region of the Eastern Cape.

The prevalence of risky alcohol use among individuals with mental illness (18.5%) reported in this study is higher than 15% in the general population of the region [29]. Likewise, the patterns of alcohol use in this study are also significantly higher than the previous report in the general population of South Africa [5] for current use (33.3% versus 33.1%) and risky alcohol use (18.5% versus 14.1%). However, a higher prevalence of risky alcohol use has been reported elsewhere, albeit using a similar definition for risky alcohol use. For instance, [42] Eberhard et al. (2015) in Sweden found a prevalence of 22% for women and 30% for men (average 26%) of risky alcohol use among individuals with mental illness in an outpatient setting, while a significantly higher prevalence of 48.5% was reported among inpatients in acute psychiatry wards in the United Kingdom [43].

The present study reports a higher prevalence of past-year use of psychoactive substance use (17.4%) in comparison to 4.1%, 4.4%, and 5.5% for the general population in the Eastern Cape, South Africa, and globally, respectively [8,44]. Several other studies have similarly reported a higher prevalence of lifetime and past-year substance use in individuals with mental illness in comparison to the general population [9,45,46,47,48].

Our study shows that the prevalence of lifetime, past-year, and risky alcohol use was higher among males than females. This trend is similar to previous studies carried out in the Eastern Cape, nationally in South Africa, and globally [29,49,50,51,52]. Males also had a higher prevalence (25.9%) of past-year use of psychoactive drugs than females (5.1%). Similar observations were made in studies done nationally in South Africa (7.9% vs. 1.3% in the previous three months), Nigeria (21.8% vs. 7.0%), the United Kingdom (12.6% vs. 6.3%), and the Netherlands [8,53,54,55]. The World Health Organisation Report [51] found a lower prevalence of lifetime abstinence from alcohol in males (34.5%) in comparison to females (54.6%). The differences in alcohol use can be attributed to the influence of gender-related socio-cultural factors, which in themselves vary between different ethnic and cultural groups [56]. In addition to the traditional gender roles and societal norms, which set the acceptable standards of behavior for both sexes, these complex dynamics influence alcohol and substance use at the population level, especially among women [50,57]. The important issue regarding differences in substance use between the sexes is that, while men are more likely to use substances and have substance use disorders, women are more susceptible to negative health and psychosocial outcomes of alcohol and drug use and face more obstacles in accessing treatment for substance use disorders [58,59,60], thus highlighting the need for prevention and intervention programs aimed specifically at this vulnerable group.

Younger individuals (18–35 years) with mental illness were more likely to engage in risky alcohol and psychoactive substance use than their older counterparts. This finding is consistent with global trends [59]. Substance use can be recreational or as a coping mechanism in difficult socioeconomic circumstances. The reasons for this are multi-factorial and can be seen at the individual, familial and societal levels. On an individual level, there is cognitive development happening in the young brain. This is a critical period that lends itself to particular vulnerability to stressors and risk-taking behaviors. Individuals might have undiagnosed, untreated mental illnesses for which they self-medicate with substances [61]. Individuals with family problems, such as ongoing conflict, physical or mental illness, and substance abuse, are also more likely to use substances as a coping mechanism. Unemployment and poverty can result in individuals using substances to escape or becoming part of the supply chain to earn money to survive [53]. Youths also use substances due to peer pressure, the need to belong, bullying (as the perpetrator or victim), and, in some instances, as a result of gang affiliations [62,63]. Therefore, younger individuals with mental illness should be categorized as a high-risk sub-population that requires targeted intervention programs at the community level.

Employed individuals with mental illness were more likely to engage in risky alcohol and psychoactive drug use in comparison to those receiving social grants. There are several plausible explanations for this result. Employed individuals have more resources at their disposal to secure alcohol and psychoactive drugs in comparison to those with social assistance (ranges from 350–1880 Rands), who tend to prioritize basic needs over substance use [64]. Individuals who are receiving social grants must have fulfilled a set of criteria that suggest that they were incapable of obtaining employment or generating income to support themselves. In addition, the results from this study also show that the majority of participants live with family members (84.9%). The money from the social assistance grants is often used to support not just the individual but also the family members with whom they live.

Interestingly, it was observed that unemployed individuals also engage in risky alcohol and psychoactive drug use in comparison to those receiving social grants. Besides the underlying mental illness precluding the patients from securing and keeping a job, it is also plausible that alcohol and psychoactive drug use impact negatively on the underlying mental illness and consequently lead to poor functioning. Evidence suggests that unemployment can be a psychosocial stressor that leads to increased use of substances by unemployed individuals [65]. Often, the source of money for procuring alcohol and psychoactive drugs remains an important question, despite being unemployed. Unemployed individuals resort to borrowing, theft, robbery, extortion, and other illegal activities in order to access substances [66]. Anecdotally in this setting, unemployed individuals share substances with friends or family members who buy them or perform occasional piece-work or resort to crime. This could be an area for future research in this population.

In this study, individuals living in urban areas were found to have a higher prevalence of both alcohol and psychoactive substance use than those living in rural areas. Similar trends have been observed in national studies in South Africa and China [5,8,67]. Individuals residing in urban areas are more likely to be employed and have better access to alcohol and psychoactive drug use. In contrast to this, studies done in first-world countries like Germany and Canada found that the use of alcohol and psychoactive substances was more prevalent in rural rather than urban areas [68,69]. The reason for the increased prevalence of risky alcohol and psychoactive substance use in the rural populations in Germany and Canada has been speculated to be due to limited access to drug education, treatment services, employment opportunities, and extra-curricular activities. These amenities are not available in rural areas [69,70].

## 5. Conclusions

This study reports a high prevalence of lifetime and past-year use of alcohol and psychoactive drug use in individuals with mental illness in the region, similar to trends seen in other international studies. These findings, therefore, highlight the need for (a) further research on this subject in this region; (b) policies and intervention programs that are aimed specifically at addressing the unique needs of this vulnerable group; (c) infrastructure (mental health facilities, substance rehabilitation centers, dual-diagnosis facilities) (d) staff to be trained in the management of this group of individuals, and; (e) screening for alcohol and substance abuse at the primary healthcare level.

## 6. Strengths and Limitations

The cross-sectional design of the study did not allow for a causal association to be established. The demographic correlates reported should therefore not be interpreted as causal factors. Self-reporting of the alcohol and psychoactive drug use, especially among women, could have been affected by social desirability bias; however, the medical records were reviewed to validate this information. Though this study was conducted in a single health facility center, it should be noted that this a referral hospital that provides mental health service to the central region of the Eastern Cape, a combined population of 1,674,637 residents. Health authorities might be able to use the findings of this study to guide and craft appropriate interventions for this vulnerable population group in the region.

## Figures and Tables

**Table 1 ijerph-18-05428-t001:** Background characteristics of study participants.

Variables	Frequency	Percent
Sex		
Male	232	59.5
Female	158	40.5
Age		
18–25	37	9.5
26–35	87	22.3
36–45	90	23.1
46–55	106	27.2
56–65	48	12.3
66 and above	22	5.6
Race		
Black	340	87.2
White	34	8.7
Coloured	14	3.6
Indian/Asian	2	0.5
Education		
No formal education	13	3.3
Grade 1–7	89	22.8
Grade 8–12	223	57.2
Tertiary	65	16.7
Marital Status		
Single	292	74.9
Married	37	9.5
Separated	25	6.4
Widowed	20	5.1
Cohabiting	16	4.1
Employment		
Employed	33	8.5
Unemployed	316	81.0
Student	11	2.8
Retired	30	7.7
Household income		
None	24	6.2
15–2000 R	114	29.2
2001–5000 R	183	46.9
5001 R and over	69	17.7
Source of income		
None	94	24.1
Job/Business	33	8.5
Social Assistance	263	67.4
Place of residence		
Rural	64	16.4
Urban	326	83.6
Living arrangement		
Living alone	19	4.9
Living with friends	18	4.6
Living family	331	84.9
Living with partner/spouse	22	5.6

**Table 2 ijerph-18-05428-t002:** Association between background characteristics and alcohol use.

Variables	Ever Consumed Alcohol			Alcohol Use in the Past Year		
	N (%)	UOR	AOR	N (%)	UOR	AOR
All	251 (64.4)			130 (33.3)		
Gender						
Male	179 (77.2)	4.03 (2.60–6.25) ***	4.50 (2.68–7.53) ***	88 (37.9)	1.69 (1.09–2.63) *	1.78 (1.08–2.92) *
Female (ref)	72 (45.6)	1	1	42 (26.6)	1	1
Age						
18–35	94 (75.8)	6.01 (3.16–11.42) ***	3.81 (1.85–7.86) ***	58 (46.8)	3.52 (1.77–6.96) ***	2.87 (1.37–6.03) *
36–45	59 (65.6)	3.65 (1.89–7.04) ***	2.48 (1.20–5.10) *	27 (30.0)	1.71 (0.82–3.59)	1.33 (0.61–2.89)
46–55	74 (69.8)	4.43 (2.33–8.44) ***	4.40 (2.14–9.04) ***	31 (29.2)	1.65 (0.81–3.40)	1.55 (0.72–3.32)
56 and above (ref)	24 (34.3)	1	1	14 (20.0)	1	1
Education						
No formal education and up to Grade 7	48 (47.1)	0.42 (0.22–0.81) *	0.29 (0.13–0.63) *	18 (17.6)	0.34 (0.17–0.70) *	0.39 (0.18–0.84) *
Grade 8–12	159 (71.3)	1.19 (0.65–2.15)	1.03 (0.52–2.16)	87 (39.0)	1.02 (0.58–1.81)	1.09 (0.58–2.03)
Tertiary (ref)	44 (67.7)	1	1	25 (38.5)	1	1
Marital Status						
Single	217 (64.4)	1.01(0.55–1.85)	1.06 (0.52–2.16)	108 (32)	0.67 (0.37–1.20)	0.67 (0.34–1.31)
Married or cohabiting (ref)	34 (64.2)	1	1	22 (41.5)	1	1
Source of income						
None	27 (81.8)	2.50 (1.45–4.30) *	1.68 (0.90–3.16)	19 (57.6)	1.97 (1.20–3.21) *	1.28 (0.75–2.20)
Job/business	206 (65.2)	2.25 (0.98–5.17)	2.11 (0.81–1.04)	98 (31.0)	3.18 (1.52–6.65) *	2.27 (1.01–5.12) *
Social assistance (ref)	9 (81.8)	1	1	7 (63.6)	1	1
Place of residence						
Rural	9 (30.0)	0.72 (0.42–1.24)	0.56 (0.30–1.04)	6 (20.0)	0.51 (0.27–0.96) *	0.47 (0.24–0.91) *
Urban (ref)	86 (62.3)	1	1	41 (29.7)	1	1

* *p*-values < 0.05; *** *p*-values < 0.001, UOR; Unadjusted odds ratio, AOR; adjusted odds ratio, ref; reference category.

**Table 3 ijerph-18-05428-t003:** Multivariable analysis showing predictors of risky alcohol use.

Variables	N (%)		
All	72 (18.5)	UOR	AOR
Gender			
Male	57 (24.6)	3.11 (1.69–5.72) ***	2.97 (1.53–5.77) *
Female (ref)	15 (9.5)	1	1
Age			
18–35	32 (25.8)	4.52 (1.67–12.23) *	3.33 (1.16–9.57) *
36–45	14 (15.6)	2.40 (0.82–7.01)	1.73 (0.57–5.30)
46–55	21 (19.8)	3.21 (1.15–8.97) *	2.76 (0.95–8.07)
56 and above (ref)	5 (7.1)	1	1
Education			
No formal education and up to Grade 7	11 (10.8)	0.86 (0.33–2.27)	0.92 (0.32–2.61)
Grade 8–12	53 (23.8)	2.22 (1.00–4.95)	2.21 (0.93–5.23)
Tertiary (ref)	8 (12.3)	1	1
Marital Status			
Single	59 (17.5)	0.65 (0.33–1.30)	0.56 (0.25–1.23)
Married or cohabiting (ref)	13 (24.5)	1	1
Source of income			
None	7 (21.2)	2.26 (1.28–3.99) *	1.68 (0.90–3.16)
Job/business	59 (18.7)	1.90 (0.80–4.51)	1.63 (0.61–4.34)
Social assistance (ref)	4 (36.4)	1	1
Place of residence			
Rural	2 (6.7)	0.41 (0.17–0.99) *	0.36 (0.14–0.90) *
Urban (ref)	26 (18.8)	1	1

* *p*-values < 0.05; *** *p*-values < 0.001, UOR; Unadjusted odds ratio, AOR; adjusted odds ratio, ref; reference category.

**Table 4 ijerph-18-05428-t004:** Substance use patterns.

Scheme	Frequencies	Percentages
Past year psychoactive substance Use	68	17.4
Inappropriate OTC/Prescription	12	3.1
Cannabis lifetime	146	37.4
Methaqualone lifetime	43	11.0
Methamphetamines lifetime	40	10.3
Cocaine Lifetime	12	3.1
Opioids lifetime	12	3.1
Hallucinogens lifetime	9	2.3
Others	8	2.1

OTC = Over the Counter.

**Table 5 ijerph-18-05428-t005:** Adjusted and unadjusted logistic regression models showing the association between background characteristics and psychoactive drug use.

Variables	Lifetime Psychoactive Drug use			Psychoactive Drug Use in the Past Year		
	N (%)	UOR	AOR	N (%)	UOR	AOR
All	155 (39.7)			68 (17.4)		
Gender						
Male	129 (55.6)	6.36 (3.88–10.42) ***	7.62 (4.24–13.70) ***	60 (25.9)	6.54 (3.03–14.12) ***	7.52 (3.17–17.82) ***
Female (ref)	26 (16.5)	1	1	8 (5.1)	1	1
Age						
18–35	69 (55.6)	8.50 (3.88–18.63) ***	5.84 (2.43–14.05) ***	39 (31.5)	4.89 (1.95–12.26) *	2.91 (1.04–8.12) *
36–45	39 (43.3)	5.18 (2.30–11.71) ***	3.98 (1.62–9.77) *	14 (15.6)	1.97 (0.71–5.41)	1.29 (0.43–2.12)
46–55	38 (35.8)	3.79 (1.69–8.47) *	3.01 (1.24–7.33) *	9 (8.5)	0.99 (0.34–2.92)	0.66 (0.21–2.12)
56 and above (ref)	9 (12.9)	1	1	6 (8.6)	1	1
Education						
No formal education and up to Grade 7	32 (31.4)	0.78 (0.41–1.50)	0.79 (0.35–1.78)	15 (14.7)	1.07 (0.44–2.62)	1.55 (0.53–4.53)
Grade 8–12	99 (44.4)	1.36 (0.77–2.41)	1.33 (0.66–2.71)	44 (19.7)	1.53 (0.70–3.33)	1.70 (0.67–4.28)
Tertiary (ref)	24 (36.9)	1	1	9 (13.8)	1	1
Marital Status						
Single	129 (38.3)	0.64 (0.36–1.15)	0.39 (0.19–0.82) *	57 (16.9)	0.78 (0.38–1.60)	0.41 (0.17–1.01)
Married or cohabiting (ref)	26 (49.1)	1	1	11 (20.8)	1	1
Source of income						
None	53 (56.4)	2.91 (1.79–4.72) ***	2.04 (1.15–3.61) *	31 (33.0)	4.49 (2.49–8.10) ***	3.16 (1.62–6.17) *
Job/business	21 (63.6)	3.93 (1.85–8.38) ***	4.66 (1.81–11.99) *	11 (33.3)	4.56 (1.99–10.45) ***	4.20 (1.57–11.25) *
Social assistance (ref)	81 (30.8)	1	1	26 (9.9)	1	1
Place of residence						
Rural	25 (39.1)	0.97 (0.56–1.67)	0.89 (0.47–1.68)	5 (7.8)	0.35 (0.14–0.92) *	0.27 (0.10–0.77) *
Urban (ref)	130 (39.9)	1	1	63 (19.3)	1	1

* *p*-values < 0.05; *** *p*-values < 0.001, UOR; Unadjusted odds ratio, AOR; adjusted odds ratio, ref; reference category.

## Data Availability

Data for this project forms part of the MMED research at Walter Sisulu University for Linda Tindimwebwa. Data included in this paper are available from Linda Tindimwebwa upon reasonable request.

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
