# Peer review of "Prevalence and Demographic Correlates of Substance Use among Adults with Mental Illness in Eastern Cape, South Africa: A Cross-Sectional Study"

_ijerph, 2021, doi:10.3390/ijerph18105428_

Round 1

Reviewer 1 Report

This is an extremely  well written an well conceived hospital-based study of substance use (especially of alcohol and cannabis but also other substances) in a population with psychiatric disorders. 

I have only minor suggestions for improvement that would strengthen the manuscript.

  1. It might be worth commenting on in Discussion the potential impact on study findings, or their generalizability, of excluding those who were a danger to self, to others or property.
  2. Line 124 ff notes a NIDA basis for alcohol consumption items but the reference (3) is to an NIAAA site on definitions alcohol measurement.  Please clarify or fix.
  3. Consent I see was in English and Xhosa but I think the translation of the instrument into Xhosa was not noted.  Also, were back-translations undertaken and if not, why not?
  4.   Line 152-156.  The one methodological mistake in measurement I see is that in the definition of risky drinking as exceeding the low risk drink limit for men is stated as 4+ (x 7 = 21 drinks weekly) and for women 3+ (x7 = 14 drinks weekly).  The values of 4+ and 3+ (men, women) are NIAAA levels of exceeding the DAILY limit, but there is separately a WEEKLY limit of < 14 for men and 7 for women.  In other words the weekly volume limit is lower than  multiplying out the daily limit.  Because of this it would be proper (unless reanalysis is possible, which would be better) to note this non standard volume level (21 drinks for men and 14 drinks for women).  This means the rate of risky drinking is quite CONSERVATIVE in this study, as presented. 
  5. There are cosmetic changes needed in the tables.  Particularly, the First columns should NOT be centered.  Rather, the heading e.g., Sex should be to the left of the response options  Male, Female.  (For all Tables.)
  6. In Table 5 fix table so 95% CIs are not on 2 lines.
  7.  

Author Response

Comment 1: It might be worth commenting on in Discussion the potential impact on study findings, or their generalizability, of excluding those who were a danger to self, to others or property.

Response: Thanks for this suggestion. We have added a sentence on this issue.

Comment 2: Line 124 ff notes a NIDA basis for alcohol consumption items but the reference (3) is to an NIAAA site on definitions alcohol measurement.  Please clarify or fix.

Response: We have corrected the reference from NIAAA to NIDA.

Comment 3: Consent I see was in English and Xhosa but I think the translation of the instrument into Xhosa was not noted.  Also, were back-translations undertaken and if not, why not?

Response: The instrument was in English, which was administered by the attending Clinicians, who are bilingual.

Comment 4: Line 152-156.  The one methodological mistake in measurement I see is that in the definition of risky drinking as exceeding the low risk drink limit for men is stated as 4+ (x 7 = 21 drinks weekly) and for women 3+ (x7 = 14 drinks weekly).  The values of 4+ and 3+ (men, women) are NIAAA levels of exceeding the DAILY limit, but there is separately a WEEKLY limit of < 14 for men and 7 for women.  In other words the weekly volume limit is lower than multiplying out the daily limit.  Because of this it would be proper (unless re-analysis is possible, which would be better) to note this non-standard volume level (21 drinks for men and 14 drinks for women).  This means the rate of risky drinking is quite CONSERVATIVE in this study, as presented. 

Response: The instrument had both definitions and patients answered either of the two categories. Risky alcohol use was then categorised as either of 4 or more and 3 or more daily drinks for male and female, respectively or 21 or more and 14 or more for weekly drinks male and female per week, respectively.

Comment 5: There are cosmetic changes needed in the tables.  Particularly, the First columns should NOT be centered.  Rather, the heading e.g., Sex should be to the left of the response options Male, Female.  (For all Tables.)

  1. In Table 5 fix table so 95% CIs are not on 2 lines.

Response: Corrections have been made.

Reviewer 2 Report

The authors have done a fine job of presenting their study. 

The ten pilot cases used to try the instrument do not seem to be sufficient to ascertain reliability and validity - I think the authors should consider writing "acceptability and feasibility" in stead.

I wondered when reading this manuscript if homebrew is a significant issue in this area. If so, the self-reported alcohol consumption is very unreliable. That does not invalidate the study, but I would like to know how the authors think about this issue.

I was also wondering if there is an issue of co-linearity. E.g., is it possible that older people or people from rural areas have very different levels of education compared with younger people or people from urban areas? I would at the very least suggest that the authors consider testing the variance inflation factor of their many co-variates.

The authors discussion of differences between countries such as China or South Africa on the one hand, and Canada or Germany on the other, is very brief and under-developed. I suspect that the authors have much more to say about this issue.

Author Response

Comment 1: The ten pilot cases used to try the instrument do not seem to be sufficient to ascertain reliability and validity - I think the authors should consider writing "acceptability and feasibility" instead.

Response: Thanks for the suggestion. Corrections have been made.

Comment 2: I wondered when reading this manuscript if homebrew is a significant issue in this area. If so, the self-reported alcohol consumption is very unreliable. That does not invalidate the study, but I would like to know how the authors think about this issue.

Response: Thanks for your insight on the locally prepared alcohol which is not a major issue in this region.

Comment 3: I was also wondering if there is an issue of co-linearity. E.g., is it possible that older people or people from rural areas have very different levels of education compared with younger people or people from urban areas? I would at the very least suggest that the authors consider testing the variance inflation factor of their many co-variates.

Response: Thanks for this comment. We have tested for collinearity using variance inflation factors, and none of the variable has more than VIF of 2, indicating there is no multi-collinearity.

Comment 4: The authors discussion of differences between countries such as China or South Africa on the one hand, and Canada or Germany on the other, is very brief and under-developed. I suspect that the authors have much more to say about this issue.

Response; We have added a sentence to provide context for the differences observed in the various countries.